# The SOCE Machinery: An Unbalanced Knowledge between Left and Right Ventricular Pathophysiology

**DOI:** 10.3390/cells11203282

**Published:** 2022-10-18

**Authors:** Jessica Sabourin, Antoine Beauvais, Rui Luo, David Montani, Jean-Pierre Benitah, Bastien Masson, Fabrice Antigny

**Affiliations:** 1Signalisation et Physiopathologie Cardiovasculaire, Inserm, Université Paris-Saclay, UMR-S 1180, 91400 Orsay, France; 2Faculté de Médecine, Université Paris-Saclay, 94270 Le Kremlin-Bicêtre, France; 3Hypertension Pulmonaire: Physiopathologie et Innovation Thérapeutique, Hôpital Marie Lannelongue, Université Paris-Saclay, Inserm, UMR-S 999, 92350 Le Plessis-Robinson, France; 4Service de Pneumologie et Soins Intensifs Respiratoires, Centre de Référence de l’Hypertension Pulmonaire, Assistance Publique-Hôpitaux de Paris (AP-HP), Hôpital Bicêtre, 94270 Le Kremlin-Bicêtre, France

**Keywords:** SOCE, SOCs, TRPCs, Orai1/3, STIM1/2, calcium signaling, left and right hypertrophy, left and right heart failure, pulmonary hypertension

## Abstract

Right ventricular failure (RVF) is the most important prognostic factor for morbidity and mortality in pulmonary arterial hypertension (PAH) or pulmonary hypertension (PH) caused by left heart diseases. However, right ventricle (RV) remodeling is understudied and not targeted by specific therapies. This can be partly explained by the lack of basic knowledge of RV remodeling. Since the physiology and hemodynamic function of the RV differ from those of the left ventricle (LV), the mechanisms of LV dysfunction cannot be generalized to that of the RV, albeit a knowledge of these being helpful to understanding RV remodeling and dysfunction. Store-operated Ca^2+^ entry (SOCE) has recently emerged to participate in the LV cardiomyocyte Ca^2+^ homeostasis and as a critical player in Ca^2+^ mishandling in a pathological context. In this paper, we highlight the current knowledge on the SOCE contribution to the LV and RV dysfunctions, as SOCE molecules are present in both compartments. he relative lack of studies on RV dysfunction indicates the necessity of further investigations, a significant challenge over the coming years.

## 1. Introduction

In pulmonary arterial hypertension (PAH), the elevated pulmonary vascular resistance due to distal pulmonary artery obstruction ultimately induces right ventricular failure (RVF) [1]. Heart failure (HF) describes the inability of a heart to maintain the workload necessary to meet oxygen and nutriment needs. The purpose of the RV is to maintain a sufficient blood flow through the lungs to achieve adequate left ventricular (LV) filling with oxygenated blood. Hence, cardiac output of the right and left heart is closely linked, as a decrease in pulmonary blood flow will decrease LV filling and, consequently, LV cardiac output. As the pulmonary circulation works at a lower pressure than the systemic circulation, the RV wall is much thinner and therefore less powerful, allowing better compliance but poorer adaptation to overload [2]. Thus, any sudden increase in pulmonary vascular resistance or blood volume may overcome the RV’s capacities, provoking a dilatation and then a decrease in stroke volume and cardiac output. If the phenomenon is chronic, the RV can, for a time, as with the LV, respond by progressive hypertrophy to improve its contractility [1,3].

Excessive RV dilatation triggers a tricuspid regurgitation favoring venous congestion, quickly assessed by clinical signs (edema, jugular distension, anasarca, and so on), biochemical markers (elevated NT-proBNP or BNP), and ultrasonography. It also eventually leads to the protrusion of the interventricular septa toward the LV, impairing its filling and contractility. These signs of RV dysfunction are critical predictors of poor prognosis, suggesting an impending HF and a high risk of death by cardiogenic shock [4]. Because pulmonary embolism [5] and pulmonary hypertension (PH) [6] increase RV afterload, they are classic cases of fatal diseases that provoke acute or chronic RVF.

Currently, treatment of RV dysfunction is based on blood volume management, using mainly diuretics to reduce RV overload while carefully avoiding systemic hypotension. In the most severe cases, inotropes and vasopressors may be needed to maintain a sustainable cardiac output [6]. None of these treatments are specific to the RV. Moreover, there is no treatment targeting the RV remodeling during chronic failure, despite being a critical factor in the severity of PH [7].

The RV is a crescent-shaped structure linked to the venous system from one side and the pulmonary circulation from the other, through the tricuspid and the pulmonary valves, respectively. Its free wall is thinner than the LV myocardium (3–5 mm versus around 1 cm) for a volume that is 10–15% greater and a mass one-third to one-sixth smaller [8,9]. Three cardiomyocyte layers are described in the LV, but only two in the RV, with myocytes arranged circumferentially on the external layer and longitudinally in the internal layer. This organization allows a peristalsis-like motion during contraction, whereas the LV systole is torsion-shaped due to oblique myocardial fibers. Notably, the RV has smaller cardiomyocytes at the histological level and contains 30% more collagen [10].

These anatomical features give the RV greater compliance, allowing the venous return variations to absorb and provide a stable stroke volume toward the low-impedance pulmonary circulation while using only one-fourth of the LV stroke work. Subsequently, the RV is more vulnerable to significant increments in pulmonary arterial pressure.

The anatomical differences between the RV and LV are the result of a different embryological origin, developed during the cardiac morphogenesis from the third to the eighth week of gestation. The LV arises from the primitive heart tube, also named the first cardiac field. In contrast, the RV is formed from the anterior part of the “second heart field”, cardiac precursor cells migrating towards the extremities of the primitive heart tube [11]. In the foetus, pulmonary vascular resistance is high; therefore, the RV and LV have equal thickness and power, producing 60% of the cardiac output [12]. This pulmonary pressure normalizes at birth, and the compliance rises as the free wall slims down. The ventricle adopts its final form within the first year.

The nourishing coronary blood flow, supplied by the right coronary artery in 80% of the population, occurs in both diastole and systole (mainly during diastole in the LV) but is more vulnerable to systemic hypotension and intracardiac pressure increase. Nevertheless, the RV’s relative resistance to ischemia is partially explained by lower oxygen consumption and higher oxygen extraction reserve at rest. Moreover, the expression of the anaerobic glycolytic enzyme is higher in RV than in LV cardiomyocytes [13]. Potential gene and protein expression differences between human LV and RV cardiomyocytes have been little explored.

The RV structure and function assessment are also more challenging than the LV. Its retrosternal position and complex geometry complicate echocardiography evaluation. Metabolic imaging is not yet feasible, as the free wall is too thin. Right-heart catheterization is the most accurate tool to measure hemodynamic features, especially to calculate pulmonary vascular resistance, but it is not a routine test. Compared to the LV, RV conductance studies are technically more challenging, partly because of the difficulty of obtaining reliable ventricular volumes without magnetic resistance imaging [2].

## 2. Ca^2+^ Signaling in the Left and Right Ventricles

The common pathophysiology of LVF and RVF is cellular remodeling, which is regulated and coincides with the remodeling of the intracellular Ca^2+^ concentration ([Ca^2+^]_i_). Ca^2+^ mishandling is a hallmark of left and right HF [14,15] and is a major cause of cardiac contractile dysfunction. Indeed, Ca^2+^ is a universal signaling ion essential for most cardiomyocyte functions, including excitation-contraction coupling (ECC) and regulation of gene expression. In mammals, controlling [Ca^2+^]_i_ is fundamental for almost all cellular processes. In the cardiac compartment, the contraction of cardiomyocytes is mainly controlled by the ECC. The depolarization during action potential (AP) activated the L-type Ca^2+^ channels (LTCC). The resulting small Ca^2+^ influx activates massive Ca^2+^ release from the sarcoplasmic reticulum (SR) through the type 2 ryanodine receptor (RyR2): a Ca^2+^-induced Ca^2+^ release (CICR) process that controls contractile myofilaments. Other processes could also participate in cardiomyocyte Ca^2+^ homeostasis, notably the store-operated Ca^2+^ entry (SOCE) as described herein.

## 3. Generality about the SOCE Machinery

This mechanism refers to the activation of the store-operated channels (SOCs) following endoplasmic/sarcoplasmic reticulum (ER/SR) Ca^2+^ depletion, an important process in modulating [Ca^2+^]_i_ in most cell types, first described by Putney in 1989 [16]. The transient receptor potential canonical (TRPC) channels were the first plasma membrane channels suspected to contribute to SOCE. TRPC channels are constituted by seven isoforms (TRPC1 to TRPC7, with TRPC2 being a pseudogene in humans), which can form homo- or hetero-tetrameric channels like TRPC1/5, TRPC1/3, TRPC1/4, TRPC3/4, TRPC4/5, and TRPC1/4/5 [17]. The inhibition of TRPC channels by pharmacological or knockdown approaches indicates that TRPC1, C4, and C5 participate to SOCE in several cell types [17,18,19,20,21,22,23].

As TRPC channels are non-selective cation (Ca^2+^, Na^+^, K^+^) channels, their electrophysiological properties do not match with the highly selective Ca^2+^ current activated by intracellular store depletion, called Ca^2+^ release-activated Ca^2+^ current (*I*_CRAC_) [24,25,26]. Under physiological conditions, the Ca^2+^ selectivity is 1000 times higher than the Na^+^ permeation. From 2005 to 2006, the two main molecular components of SOCE and the *I*_CRAC_ were identified by several teams: the STIM (stromal interaction molecule) protein, which is an ER/SR Ca^2+^ sensor with its N-terminal EF-hand domain; and Orai (also named CRACM), which is the highly Ca^2+^ selective for typical SOCs. Indeed, contrary to TRPC channels, the electrophysiological properties of the Orai1 channel perfectly fit with the native *I*_CRAC_, a sustained non-voltage-activated Ca^2+^ inward current with an inward rectification. There are two homologous STIM proteins, called STIM1 and STIM2, each having different variants that have been characterized since the discovery of the protein family. STIM1 has two variants: STIM1 long (STIM1L), which is associated with rapid activation of SOCE only in skeletal myotubes through permanent cluster formation at the plasma membrane with Orai1 [27]; and STIM1B, a neuron-specific variant [28]. The STIM1 homolog, STIM2, has also 2 variants: STIM2.1 (or STIM2β), which is known to have inhibitory activity on SOCE; and the STIM2.2 (or STIM2α), a SOCE stimulator variant [29]. The Orai family comprises three proteins encoded by independent genes: Orai1, Orai2, and Orai3. In most cells studied so far, STIM1 and Orai1 constitute the core machinery of the ubiquitous SOCE pathway.

Following the molecular identification, the SOCE activation mechanism was explained by several studies mainly performed in non-excitable cells. A decrease in the ER/SR Ca^2+^ concentration induces a conformational change that extends the STIM1 cytosolic tail, which then promotes its oligomerization and exposes the CAD/SOAR (CRAC activation domain/STIM-Orai activating region) domain and the lysine-rich tail of STIM1. At the same time, the exposure of the CAD/SOAR domain allows STIM1 to trap Orai1 channels via interactions with their C-terminal cytosolic domain [30,31,32,33,34]. 

While STIM1/2 constitutes the main regulatory protein for Orai1, mechanisms leading to TRPC channels’ activation are still being investigated. The activation of TRPC channels predominantly occurs downstream of the GPCR-G_q_-PLC and receptor tyrosine kinases, coupled to PLCγ via IP_3_-dependent Ca^2+^ release from the ER or DAG production. Ca^2+^ itself, via calmodulin and other Ca^2+^-binding proteins or PIP_2,_ are also important regulators of TRPC channels. These multiple mechanisms of TRPC channels’ activation possibly depend on their subunit composition or signaling complex environment. Nonetheless, it has been also reported that STIM1 directly or indirectly interacts with TRPC channels (all except TRPC7), conferring them a SOCE function [35,36]. Moreover, the group of Ambudkar demonstrated that TRPC1 function could also depend on the Orai1-mediated Ca^2+^ microdomain, which generates the recruitment of TRPC1 into the plasma membrane, which is activated by STIM1. 

## 4. The SOCE in the Ventricles 

Since Putney’s work, the existence of SOCE in non-excitable cells has been well known [37]. In 2002, it was reported, for the first time, the presence of SOCE activated by passive and active Ca^2+^ depletion in neonatal rat cardiomyocytes [38] and embryonic and neonatal mice myocytes [39]. The SOCE was also shown, in neonatal rabbit cardiomyocytes, to regulate the SR Ca^2+^ load [40]. The presence of this Ca^2+^ entry has been subsequently repeatedly suggested to be a feature of an immature cardiac system.

In adult mouse cardiomyocytes, SOCE, measured by fluorimetry and patch-clamp, is also detected following the passive SR Ca^2+^ depletion [41]. However, the huge SOCE amplitude inducing hypercontracture in this report is highly questionable. More recently, Wen and colleagues demonstrated that complete SR Ca^2+^ depletion is essential to optimally activate moderated SOCE in adult mouse cardiomyocytes [42]. Indeed, it is now widely accepted that the SOCE amplitude in adults is significantly lower than in embryonic and neonatal cardiomyocytes and declines during development [39,43,44,45,46,47,48,49,50,51]. Moreover, it has been repeatedly reported that only 20–70% of adult cardiomyocytes exhibit SOCE in response to SR Ca^2+^ depletion [43,48,49,52]. Nonetheless, many reports did not use the appropriate and optimal SR Ca^2+^ depletion protocol to induce sufficient Ca^2+^ emptying to activate SOCs [43,44]. For example, Wu and colleagues show that only 20% of the cells present SOCE, but with a protocol mobilizing only inositol trisphosphate receptors (IP_3_R) and not the RyR2 [43]. In early studies, evidence of SOCE is also suggested using pharmacological inhibitors, such as 2-APB, SKF-96365, BTP2, or inorganic cations, such as Gd^2+^ and La^3+^, whose specificity is controversial. Lately, the existence of SOCE in the cardiac tissue was confirmed by the detection of the different isoforms of TRPC1/C3-C7, Orai1–3 channels, and STIM1–2 proteins in the embryonic, neonatal, and adult cardiomyocytes of rats, mice, cats, zebrafish, squirrels, and chickens, as well as in human left myocardial tissue, with an expression that declines after birth [27,40,44,46,51,53,54,55,56,57,58,59,60,61,62,63,64,65,66,67,68].

## 5. The SOCE in the Adverse LV Remodeling

### 5.1. Role of STIM1

By manipulating their expression, several in vivo studies have demonstrated the active participation of cardiac STIM1/Orai1-mediated SOCE during hypertrophy and HF. 

For STIM1, the changes in its expression vary according to the experimental pathological models (Table 1 and Figure 1). The first study by Ohba et al. in 2009 showed no change in expression of STIM1 in a model of compensated hypertrophy by abdominal aortic banding constriction (AAB model) over 4 weeks in rats [57]. Later, in the same experimental model, upregulation of STIM1 with the doubling of SOC current (*I*_SOC_) is observed after 14 and 28 days of pressure overload [69]. A second current, independent of the SR Ca^2+^ load and activated very rapidly, with an inward rectification reflecting an Orai1-mediated current, is also observed under this hypertrophic stress condition. This suggests that STIM1 can activate Orai1 and non-selective cationic channels, like TRPC channels, in adult cardiomyocytes. The rapid activation of this current can be also explained by the constitutive association between STIM1 or STIM1L and Orai1 at the membrane, as observed in sinoatrial node cells [70]. Luo et al. also demonstrated an increase in STIM1L variant expression, associated with increased SOCE after compensated hypertrophy induced by constriction of the thoracic aorta for 3 weeks in mice, but they did not find any increase in the classic STIM1 variant [46]. After 2 or 4 weeks of pressure overload in mice by the transverse aorta constriction (TAC model), inducing compensated or decompensated hypertrophy, an increase in STIM1 expression is found [61,71]. By contrast, we and others found no changes in STIM1 expression after 2, 5 or 8 weeks of TAC in mice [72,73]. Correll et al. chose to use a mouse model of cardiac specific STIM1 overexpression, thereby mimicking its upregulation, to study its role in the pathology. Under basal conditions, the transgenic STIM1 mice show sudden cardiac death at 6 weeks. The surviving mice develop severe HF, associated with induction of the fetal gene program, histopathology, and mitochondrial structural alterations, at 12 weeks of age [61]. More recently, adult young cats, with compensated hypertrophy induced by ascending aorta banding for 4 months, show increased protein expression of STIM1 [74]. Lastly, after neurohormonal stimulation by Angiotensin II (AngII) over 4 weeks, STIM1 expression is also raised [75]. In human, STIM1 mRNA level is also significantly increased in LVF [60]. The novel STIM2.1 variant is detected in both healthy and failing human myocardium. In severe LVF, STIM2.1 expression is significantly decreased. The lower ratio of STIM2.1/STIM2.2 ratio in failing hearts indicates a shift from the SOCE-inhibiting STIM2.1 variant to the stimulatory one, STIM2.2, suggesting an exacerbated SOCE in failing patients.

At baseline, the cardiac specific deletion or overexpression of STIM1 in mice leads to a deterioration of the cardiac function with age [68,76,77,78], demonstrating that the basal level of STIM1 is critical for maintaining normal LV function. However, interestingly, STIM1 silencing appears to protect the animals subjected to pressure overload against the associated hypertrophic development. Indeed, in vivo STIM1 silencing by Ad-shRNA decreases STIM1-dependent *I*_SOC_ and protects the heart from pressure overload-induced hypertrophy by reducing signaling pathway activation such as Calcineurin (CaN)/NFATc3 [69]. After induction of compensated hypertrophy in mice by AAB for 4 weeks, in addition to STIM1 and Orai1 upregulation, a decrease in the SARAF (SOCE-associated regulatory factor) protein expression has also been reported, which inhibits STIM1-Orai1 interaction, whereas SARAF overexpression prevents STIM1 and Orai1 upregulation and attenuates hypertrophic development [79]. *Stim1*-KO mice, after 5 weeks of TAC, exhibit preserved left ventricular mass and cardiac function associated with normal extracellular *signal*-regulated kinase 1 and 2 (ERK1/2) activity and CaN expression [77]. STIM1 silencing via AAV9 before induction of pressure overload by TAC in mice promotes the transition from adaptive hypertrophy to maladaptive hypertrophy with a dilated failing heart and the presence of significant pulmonary edema [76]. When STIM1 extinction occurs 3 weeks after the establishment of pressure overload-induced cardiac hypertrophy, mice also exhibit marked ventricular dilatation associated with systolic dysfunction compared to WT mice, reflecting a more rapid transition to HF. STIM1 would, therefore, be a key element in the development and the persistence of compensated cardiac hypertrophy to preserve the transition to HF. The chronic treatment with the cholinesterase inhibitor, pyridostigmine, after AAB or TAC in rats or mice, respectively, attenuates cardiac hypertrophy and improves cardiac contractile performance and rhythmic activity by suppressing the interaction of Orai1/STIM1 [80] or diminishing pathologically enhanced STIM1 expression and SOCE in hypertrophied myocytes [71].

Finally, the cardiac STIM1 overexpressing mice subjected to pressure overload by TAC or to chronic neurohormonal stimulation by infusion of AngII/Phenylephrine (PE) or isoproterenol for 2 weeks demonstrated severe maladaptive hypertrophy, with a decrease in systolic function and the appearance of pulmonary edema, demonstrating a deleterious effect of STIM1 overexpression on cardiac function under chronic stress [61]. 

With regard to the various in vivo experimental models of cardiac hypertrophy, there are differences as to whether or not STIM1 expression is increased. Regardless, altering STIM1-dependent signaling in vivo is deleterious for cardiac function, supporting the idea that preserving STIM1 expression and activity under physiological and pathological conditions is instrumental. On the other hand, most of the studies have investigated the role of STIM1 in the pathogenesis of compensated hypertrophy. It would be interesting to explore its role at more severe stages, where systolic dysfunction is established.

### 5.2. Role of Orai Channels

In line with the previous observations, mRNA and protein levels of Orai1 are found to be upregulated in mouse LV hypertrophy induced by pressure and volume overloads, such as 4 days or 5 weeks after TAC, 4 weeks after AAB, 2 weeks after AngII infusion, 4 days post-myocardial infarction, or after ischemia-reperfusion injury [64,72,79,81,82,83] (Table 1 and Figure 1). An increased expression of Orai3 is also found after compensated or decompensated hypertrophy in mice and cats [72,74,81]. In the end-stage human LVF tissue, Orai1 expression is decreased by 30% only in males. Orai1 expression is unchanged in females with LVF, suggesting the possible cardioprotective effects of estrogen by maintaining stable Orai1 expression [62]. In contrast, enhanced Orai1 expression and increased SOCE are found in fibroblasts from end-stage human LVF patients, associated with an enhanced collagen secretion capacity [84]. No changes are detected for Orai2 and Orai3 in humans (Table 1).

Interestingly, the effect of Orai1 invalidation at baseline varies between species. In zebrafish, Orai1 deficient embryos spontaneously develop severe HF and bradycardia [64]. Similarly, heart-specific suppression of *Orai1* in larvae and adult drosophila resulted in reduced contractility that is consistent with dilated cardiomyopathy [84]. However, Orai1-deficient or nonfunctional Orai1 mice have normal cardiac function, unlike those with a STIM1 deficiency [72,73,81]. Later, in drosophila, it was found that heart-specific genetic upregulation of constitutive active STIM1 and Orai1 proteins in larvae and adults resulted in significant hypertrophy [85].

The Orai1-deficient mice (*orai1^+/−^*) subjected to pressure overload (TAC) for 8 weeks have decreased survival, with faster cardiac functional loss and more significant ventricular dilation when compared to WT mice [73]. This suggests that the loss of Orai1 accelerates the pathology and results in the more raid development of dilated cardiomyopathy and HF. Cardiac hypertrophy level is similar between groups. However, *orai1^+/−^* mice are no longer able to compensate for the overload, leading to the development of more severe systolic dysfunction. This can be explained by significant apoptosis without differences in hypertrophic and fibrotic markers.

In a model of HF with preserved ejection fraction by AngII infusion for 2 weeks in mice, the cardiomyocyte-specific and temporally inducible Orai1 knockout mouse line (Orai1^CM-KO^) show a slight decrease in systolic function with increased fibrosis compared to WT mice, suggesting a worsening of the pathology and a transition towards maladaptive hypertrophy. However, given the experimental echocardiographic data, these conclusions seem hasty [81]. In contrast, we recently demonstrated that, after chronic pressure overload (5 weeks post-TAC), cardiomyocyte-specific dominant negative (dn)Orai1^R91W^ (loss of function Orai1 mutation) or in vivo pharmacological selective inhibition of Orai1 protects mice against LV systolic dysfunction and fibrosis, without a change in LV hypertrophy. The protection against LV dysfunction is associated with a normalization of SOCE, [Ca^2+^]_i_ transients amplitude, SR Ca^2+^ load, cardiomyocyte contractility, and sarco/endoplasmic reticulum Ca^2+^-ATPase (SERCA2a) expression, in which the protein tyrosine kinase (Pyk2)/mitogen-activated protein kinase (MEK)/ERK cascade is, in part, involved [72]. 

Saliba et al. showed the involvement of the Orai3 isoform during compensated cardiac hypertrophy induced by pressure overload in rats (4 weeks after AAB) [63]. In hypertrophied rats, the authors observed an increase in the interaction of Orai3 with STIM1/Orai1, leading to an increased rate of Orai3-mediated Ca^2+^ entry. This voltage-independent Ca^2+^ entry is not involved in the cardiac ECC, since the amplitude and the relaxation constant of the Ca^2+^ transients are not modified in hypertrophied rats in the presence of siRNAs directed against Orai3 or Orai1. Moreover, during pathology, arachidonic acid current (*I*_ARC_) mediated by Orai3 is increased [63]. However, the authors did not address the functional significance of this enhanced Orai3-mediated Ca^2+^ entry in the pathological states [63]. Using constitutive and inducible cardiomyocyte-specific Orai3 knockout (KO) (Orai3^cKO^) mice, Gammons et al. found that cardiac Orai3 deficiency lead to LV dysfunction progressing to dilated cardiomyopathy and LVF. Cardiomyocytes isolated from Orai3^cKO^ mice exhibit profoundly altered myocardial Ca^2+^ cycling and mitochondrial morphology. A more dramatic cardiac phenotype emerged when Orai3 is removed in adult mice using a tamoxifen-inducible Orai3^cKO^ mouse, indicating that the loss of cardiac Orai3 is critical for LV function. In a pathological state, Orai3^cKO^ mice subjected to pressure overload (2 weeks post-TAC) developed a fulminant dilated cardiomyopathy with rapid HF, characterized by interstitial fibrosis and apoptosis [86]. This has also been confirmed in vivo in two models of adaptive hypertrophy by infusion of isoproterenol for 15 days or by AAB for 28 days in rats. By intramyocardial injection of siRNA directed against Orai3, a rapid transition to maladaptive hypertrophy is demonstrated, showing a deleterious effect of knockdown Orai3 during adaptive hypertrophy [87].

In contrast to Orai3, the studies carried out in vivo on the role of Orai1 in hypertrophy and HF remains, to date, quite contradictory, probably due to the different experimental pathological models. Despite these discrepancies, all studies support the idea of a significant role of Orai1-mediated SOCE in the development of hypertrophy and HF. 

### 5.3. Role of TRPC Channels

Under physiological conditions, all TRPCs-deficient mice presented no cardiac abnormality compared with WT mice [82], indicating that TRPC channels are not crucial for maintaining normal LV function. However, evidence suggested that TRPC channels contribute to the pathogenesis of LV hypertrophy or failure.

Almost all TRPC isoforms are differentially expressed in left ventricular tissue and cells isolated from experimental animal models of LV remodeling and dysfunction, and in myocardial biopsies from failing human hearts (Table 1 and Figure 1). Indeed, in different in vivo experimental rodent models of hypertrophy and HF induced by pressure or volume overload (abdominal, ascending or transverse aortic constriction, Dahl salt-sensitive, spontaneously hypertensive (SHR), myocardial infraction, ischemia-reperfusion injury) or by neurohormonal stress (PE, Endothelin-1 (ET-1), AngII, Isoproterenol), an increased expression of TRPC1 [54,57,59,65,82,88,89,90,91,92], TRPC2 [90], TRPC3 [59,90,93,94,95,96], TRPC4 [59,81,97], TRPC5 [83] and TRPC6 [59,72,81,90,94,95,96,98,99,100,101,102,103,104] has been reported. Additionally, increased TRPC1 and TRPC5 expression in failing human hearts [93,105,106], increased TRPC6 mRNA in human LV with dilated cardiomyopathy [100], and decreased expression of TRPC4 in isolated LV myocytes from patients with ischemic cardiomyopathy and severe HF have been shown [105,107] (Table 1).

**Table 1 cells-11-03282-t001:** SOCE Molecules Expression after Hypertrophic Stresses of the Left Ventricle. ↑: increase; ↔: no change: ↓ decrease; ND: not determined.

SOCE Molecules	Expression Level	SOCE Function	Species	Induction of the LV Remodeling	References
STIM1L	↑ (mRNA/Protein)	↑ SOCE	Mouse	Thoracic aortic banding (3 weeks)	[46]
STIM1	↔ (Protein)				
STIM2	↔ (mRNA)				
TRPC6	↑ (mRNA)	ND	Mouse	Thoracic aortic banding (3 weeks)/Calcineurin Tg mouse	[100]
			Human	Dilated cardiomyopathy	
TRPC1, TRPC3, TRPC4	↔ (mRNA)		Mouse	Calcineurin Tg mouse	
TRPC3	↑ (Protein)	ND	Rat	Isoproterenol (4 mg/kg/day, 4 days)	[93]
				Spontaneous hypertensive heart failure (SHHR, 19 months)	
			Mouse	Thoracic aortic banding (7 days)	
				Calcineurin Tg mouse (2 months)	
TRPC5	↑ (mRNA/Protein)	ND	Human	Idiopathic dilated cardiomyopathy	
TRPC1, TRPC4, TRPC6	↔ (mRNA/Protein)				
STIM1	↑ (mRNA)	ND	Human	Severe LV heart failure (NYHA III-IV class)	[60]
Orai1	↔ (mRNA in female) ↓ (mRNA in male)				
STIM2, Orai2, Orai3	↔ (mRNA)				
TRPC4	↓ (mRNA)	ND	Human	Ischemic cardiomyopathy	[107]
TRPC1	↑ (mRNA)	ND	Human	Hypertrophic cardiomyopathy, Heart failure	[106]
TRPC1, TRPC5	↑ (mRNA)	ND	Human	End-stage heart failure (NYHA III-IV class)	[105]
TRPC3, TRPC6	↔ (mRNA)				
TRPC4	↓ (mRNA)				
Orai1, TRPC5	↑ (mRNA/Protein)	↑ SOCE	Rat	Ischemia (45 min)/Reperfusion (1 week) injury	[83]
STIM1, Orai1, TRPC1	↑ (mRNA/Protein)	ND	Mouse	Ischemia (30 min)/Reperfusion (24 h) injury	[82]
STIM1, Orai1	↑ (mRNA/Protein)	ND	Mouse	Abdominal aortic banding (4 weeks)	[79]
TRPC1	↑ (mRNA/Protein)	ND	Rat	Abdominal aortic banding (4 weeks)	[57]
STIM1	↔ (mRNA/Protein)				
TRPC5, TRPC6	↔ (Protein)				
TRPC1	↑ (mRNA/Protein)	ND	Rat	Abdominal aortic banding (4 weeks)	[92]
TRPC3, TRPC5, TRPC6	↔ (mRNA/Protein)				
STIM1	↑ (Protein)	↑ *I*_SOC_ and *I*_CRAC_	Rat	Abdominal aortic banding (14 days and 28 days)	[69]
STIM1	↑ (mRNA/Protein)	ND	Cat	Ascending aorta banding (4 months)	[74]
Orai3	↑ (mRNA)				
Orai1, STIM2	↔ (mRNA/Protein)				
TRPC6	↑ (Protein)	ND	Mouse	Ascending aortic banding (6 weeks)	[103]
TRPC1	↔ (Protein)				
Orai1	↑ (mRNA/Protein)	ND	Mouse	Transverse aortic banding (4 days) Myocardial infarction (4 days)	[64]
STIM1	↑ (Protein)	ND	Mouse	Transverse aortic banding (2 weeks)	[61]
STIM1, Orai1, Orai3	↑ interaction between Orai3 and STIM1/Orai1 (Protein)	↑ Orai3-mediated Ca^2+^ entry and ↑ *I*_ARC_	Rat	Transverse aortic banding (4 weeks)	[63]
STIM1, Orai1, Orai3	↔ (Protein)				
TRPC3, TRPC6	↑ (mRNA)	ND	Mouse	Transverse aortic banding (7 days)	[94]
TRPC6	↑ (Protein)				
TRPC1	↔ (mRNA)				
TRPC6	↑ (mRNA)	ND	Mouse	Transverse aortic banding (8 weeks)	[104]
TRPC1	↑ (Protein)	ND	Mouse	Transverse aortic banding (4 weeks)	[65]
TRPC3, TRPC4, TRPC6, STIM2	↔ (Protein)				
Orai1, Orai3, TRPC6, STIM2	↑ (mRNA/Protein)	↑ SOCE	Mouse	Transverse aortic banding (5 weeks)	[72]
TRPC1, TRPC3, TRPC4, TRPC5, STIM1, Orai2	↔ (mRNA/Protein)				
TRPC1	↑ (Protein)	↑ *I*_SOC_	Mouse	Transverse aortic banding (4, 8 weeks)	[54]
TRPC3, TRPC6	↔ (Protein)				
STIM1	↑ (Protein)	ND	Mouse	Transverse aortic banding (28 days)	[71]
TRPC4α, TRPC4β	↑ (Protein)	ND	Mouse	Transverse aortic banding (8 weeks)	[97]
Orai1, Orai2, Orai3, STIM1, TRPC4, TRPC6	↑ (mRNA)	ND	Mouse	Angiontensin II infusion (3 mg/kg/day, 2 weeks)	[81]
TRPC1, TRPC3, STIM2	↔ (mRNA)				
STIM1	↑ (mRNA/Protein)	ND	Mouse	Angiontensin II infusion (400 ng/kg/min, 4 weeks)	[75]
TRPC1, TRPC3, TRPC4, TRPC6	↑ (mRNA)	↑ SOCE	Mouse	Myocardial infarction (1, 2, 6 weeks)	[59]
TRPC2, TRPC5	↔ (mRNA)				
TRPC6	↑ (mRNA/Protein)	ND	Rat	Myocardial infarction (1 month)	[95]
TRPC3, TRPC6	↑ (Protein)	ND	Rat	Myocardial infarction (1, 6, 24 h)	[96]
TRPC1	↑ (Protein)	↑ Strech-activated Ca^2+^ entry	Rat	Isoproterenol (5 mg/kg/5 days, 5 weeks)	[88]
TRPC3, TRPC6	↔ (Protein)				
TRPC6	↑ (mRNA)	*↑ I* _SOC_	Mouse	Isoproterenol (2 mg/kg/day, 10 days)	[101]
TRPC1	↑ (mRNA/Protein)	ND	Rat	Spontaneously hypertensive rat (SHR)	[89]
TRPC6	↔ (mRNA) ↑ (Protein)	ND	Rat	Spontaneously hypertensive rat (SHR)	[102]
TRPC1	↑ (Protein)	↑ Strech-activated Ca^2+^ entry	Mouse	*mdx* with dilated cardiomyopathy (9–12 months)	[91]
TRPC1, TRPC6	↑ (mRNA/Protein)	ND	Mouse	Tg dn-NRSF (neuron-restrictive silencer factor, 12–16 weeks)	[90]
TRPC2	↑ (mRNA)				
TRPC3	↓ (mRNA), ↑ (Protein)				

This TRPCs upregulation leads to exacerbated and cumulative SOCE and/or receptor-operated Ca^2+^ entry (ROCE) that activate signaling pathways, such as CaN/NFAT [43,44,89,94,98,101,103,108,109] and/or Calmoduline Kinase II [59,110], but also the NF-κB pathway [106], cascades involved in the process of hypertrophic remodeling, in particular, in the re-expression of fetal genes (MYH7, ACTA) and the induction of pro-hypertrophic genes (NPPA, NPPB, MEF2a, etc.).

The use of KO mouse models or models harboring dominant-negative TRPC channels mutations and pharmacological tools has confirmed their importance in hypertrophic development. Indeed, KO mice for *trpc1* [54,92], double *trpc1/trpc4* KOs [111,112], *trpc3* KO [113,114,115], *trpc6* KO [116], double *trpc3/trpc6* KOs [117], dn-TRPC3, dn-TRPC4 and dn-TRPC6 [43,54,111] are protected from adaptive and maladaptive hypertrophy induced by mechanical stress or neurohormonal stimulation by decreasing the activity of the CaN/NFAT pathway [43,111,118], and/or interstitial fibrosis [43,111,113], or the expression of the Cav1.2 [115] or by alleviating cardiac mitochondrial dysfunction [114]. These findings highlight the cardio-protective effect of TRPCs’ downregulation against cardiac hypertrophy and failure.

On the other hand, the specific TRPC3 inhibitor, Pyr3, attenuates the adaptive hypertrophy induced by pressure overload in mice, once again supporting the significant role of TRPC3 in cardiac hypertrophy [119]. The combined pharmacological inhibition of TRPC3 and TRPC6 by selective antagonists GSK2332255B and GSK2833503A in vitro and in vivo is also cardioprotective, but with a limited effect in vivo [117]. Finally, selective in vivo inhibition of TRPC6 by BI749327 also improves cardiac function and prevents fibrosis deposition in mice subjected to pressure overload [104]. All the studies mentioned above show the significant role of TRPC channels in the development and progression of hypertrophy and HF. TRPCs-mediated SOCE/ROCE inhibition appears to be beneficial during the pathology. However, an individual modulation of the TRPC channels’ activity does not exist and is still challenging to achieve, because of the complexity of the TRPCs hetero-tetramerization.

### 5.4. The SOCE in the Adverse RV Remodeling

Contrary to the LV, the SOCE machinery expression in the RV is not yet studied and the physiological role of TRPC/Orai1/STIM1 proteins in adult RV cardiomyocytes is poorly investigated. In the RV, only a few pioneer studies demonstrated the contribution of SOCs in RV remodeling and dysfunction. Benoist et al. showed that *Trpc1* mRNA expression is decreased and *Trpc6* mRNA expression is increased in RVF induced by monocrotaline (MCT) exposure in rats (Table 2). This dysregulation could contribute to arrhythmias in this experimental model [120]. In maladaptive RV hypertrophy, induced by MCT exposure in rats, we found an increased expression of the glycosylated form of Orai1, as well as upregulation of TRPC1/TRPC4 expression and of the long STIM1 (STIM1L), associated with a decrease in the classic STIM1 protein in hypertrophied RV cardiomyocytes. This is correlated with an increased *I*_SOC_ current that is responsible for higher SR Ca^2+^ content and Ca^2+^ cycling, suggesting their contribution to the RV remodeling during hypertrophic stress [121].

### 5.5. Available Pharmacological Tools to Target SOCE in Heart Failure

Since the identification of Orai1 as the SOCs archetype, Orai1 is identified as a key player in several pathological contexts. That is why the arsenal of pharmacological Orai1 inhibitors has significantly increased over the last decade. Indeed, there is a well-described family of pyrazole derivatives, with the commonly used BTP2 (N-{4-[3,5-bis(Trifluoromethyl)-1 H-pyrazol-1-yl]phenyl}-4-methyl-1,2,3-thiadiazole-5-carboxamide or YM-58483) [122,123,124,125] and GSK compounds (GSK7975A and GSK-5503A) [126], the BTP2-derivative Synta-66, which binds directly Orai1 [127,128,129] and its suitable in vivo and more selective analog JPIII that we have developed [72,130]. The less-used AnCoA4 [131] and 5J4 (N-[[(6-Hydroxy-1-naphthalenyl)amino]thioxomethyl]-2-furancarboxamide) [132] also exist. All these compounds have never been tested in humans. However, some Orai1 blockers (RP318, PRCL02, CM2489, Auxora) have already been tested in clinical trials (phase I/IIa) for severe plaque psoriasis, acute or chronic pancreatitis, and COVID-19 pneumonia [133,134,135,136], where all are well tolerated in patients [136].

We have demonstrated that Orai1 function contributes to human pulmonary arterial hypertension (PAH) and to several preclinical models of pulmonary hypertension (PH). We have showed that Orai1 inhibition, by BTP2, 5J4, or JPIII, corrects in vitro the aberrant phenotypes of human pulmonary artery smooth muscles cells and attenuates in vivo PH in rat models, suggesting that Orai1 inhibition should be considered a relevant therapy in PAH [130]. We have also demonstrated that, after chronic pressure overload, JPIII markedly improves the left ventricular systolic function and Ca^2+^ handling by preventing the Ca^2+^ cycling mishandling, SERCA2a downregulation and fibrosis, without causing adverse effect. Our findings suggest that Orai1 inhibition has a potential favorable hemodynamic value to protect the heart from maladaptive hypertrophy and might represent a new inotropic support to help to relieve systolic dysfunction [72]. Taken together, Orai1-mediated SOCE may be a novel therapeutic target to consider in PAH and HF. It would be interesting to test the Orai1 inhibitors used in clinical trials in PAH and HF, especially since good safety in humans has already been demonstrated in several clinical trials.

Contrary to Orai1 blockers, existing TRPC channel inhibitors are not selective of all TRPC channel isoforms. For example, the xanthine-based inhibitors inhibit TRPC1, TRPC5, and TRPC6 at nanomolar concentrations [137,138]. The HC-608 compound, also called Pico145, is described as one of the most potent inhibitors of TRPC1, TRPC4, and TRPC5 [139], while HC-070 is described as an inhibitor of TRPC4 and TRPC5 [140]. The ML-204 (4-Methyl-2-(1-piperidinyl)quinoline) compound is reported as a selective TRPC4 inhibitor [138], and AC1903 (C19H17N3O) as a selective TRPC5 inhibitor [141], but they are also described to inhibit TRPC4 and TRPC5. Alternatively, GFB-8438, a derivate of pyridazinone 1 (Pyr1), is a potent TRPC4 and TRPC4/C5 inhibitor [142,143], while Pyr3 and Pyr10 are recognized as TRPC3 inhibitors [119]. The GSK2833503 compound is also a potent inhibitor of TRPC3 and TRPC6 [117,144]. Several compounds, including BI749327, SAR7334 SH045, and AM-1473, are developed to specifically inhibit TRPC6 [104,144,145,146]. Finally, the TRPC5 channel inhibitor, GFB887, is currently the only molecule tested in clinical Phase I on healthy volunteers and now in Phase II for patients with diabetic nephropathy or with focal segmental glomerulosclerosis [147].

Some of them can discriminate each TRPC isoform in vitro by using specific concentrations, but their use in vivo is challenging, due to their lack of selectivity at higher concentrations. Moreover, some compounds are not stable in vivo, which prevents their use in preclinical or clinical research. The development of selective TRPC channel inhibitors is needed and should be accelerated by recent cryo-EM structure discovery [148,149]. Developing more selective pharmacological tools to selectively inhibit each TRPC isoform could facilitate the emergence of innovative therapies for multiple diseases, such as LVF and RVF.

## 6. Conclusions and Future Directions

RVF is the most common cause of death in patients with PH, as the RV rapidly switches from adaptive to maladaptive hypertrophy in contrast to the LV. Although stopping or slowing down the progression of the RV disease appears necessary to avoid double transplantation, no treatment explicitly addresses RV dysfunction. As summarized in this review, SOCs are crucial for Ca^2+^ mishandling in LV dysfunction and could also be involved in RV dysfunction. Nevertheless, few studies have investigated the role of SOCs in the development of RV dysfunction. Additionally, the existing studies were performed in small animal models of RV dysfunction caused by pulmonary vessel remodeling. Further basic research studies are needed to decrypt the contribution of SOCE in RV remodeling in a large animal model or pulmonary arterial banding animals, and to investigate SOCs as novel potential therapeutic options during RV dysfunction.

## Figures and Tables

**Figure 1 cells-11-03282-f001:**
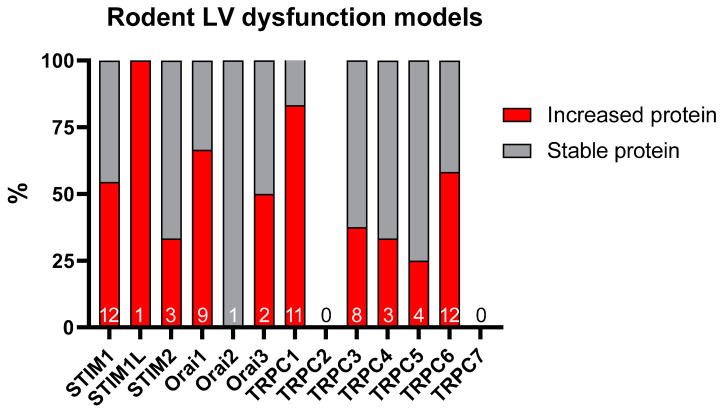
Illustration of the changes in protein expression of the SOCE machinery in rodent left dysfunction models. *n* = number of articles.

**Table 2 cells-11-03282-t002:** SOCE molecules expression after hypertrophic stresses of the Right Ventricle. ↑: increase; ↔: no change ↓ decrease; ND: not determined.

SOCE Molecules	Expression Level	SOCE Function	Species	Induction of the RV Remodeling	References
TRPC1	↓ (mRNA)	ND	Rat	Monocrotaline (60–80 mg/kg, 3–5 weeks)	[120]
TRPC6	↑ (mRNA)				
TRPC1, TRPC4, glycosylated Orai1, STIM1L	↑ (Protein)	↑ *I*_SOC_	Rat	Monocrotaline (60 mg/kg, 3 weeks)	[121]
STIM2, Orai3, TRPC3, TRPC6	↔ (Protein)				
TRPC5, STIM1	↓ (Protein)				

## Data Availability

Not applicable.

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
