# Peer review of "The SOCE Machinery: An Unbalanced Knowledge between Left and Right Ventricular Pathophysiology"

_cells, 2022, doi:10.3390/cells11203282_

Round 1
Reviewer 1 Report
This review by Sabourin et al. provides a well written overview on current knowledge about the cardiac SOCE machinery/pathway and its critically role in cardiac remodeling. The authors address a highly relevant and interesting topic by highlighting the problem of missing information about this pathway for the right ventricle (RV). The review starts out with a very nice elaboration of known structural and functional differences between ventricles and continues a summary on SOCE aspects that probably apply in general to various (all) tissues and may be similar in in RV and left ventricles (LV). As it stands the authors provide overall a general summary on SOCE mechanisms and their potential relevance for cardiac remodeling.
Major points:
The issue of what is known and what is unknown about cardiac SOCE heterogeneity, specifically differences between cardiac ventricles, and why one might indeed anticipate such differences, is then largely missing in rest of the manuscript. For instance the authors state that all TRPCs are expressed in "ventricular tissues". What about a comparison of expression between RV vs LV, are there indeed no reports available? Only in the end there is a rather brief paragraph on “knowns” for the potential role of SOCE in the right ventricle.
I would suggest to include respective statements throughout the manuscript to provide the reader with a bit more information as to what is known, indeed unknown and/or what is expected in terms of differences between RV and LV SOCE. I would like to see a bit more background why one actually would expect substantial heterogeneity in cardiac SOCE.
There is another aspect I have a particular problem with:
In their outline of general aspects of the SOCE machinery the authors seem to take a rather "historic approach" starting out with posing TRPC channels as a “store-operated channel family”. This reflects a simple historic point of view based on the initial suspicion that TRPCs might be “store-operated” in the sense of featuring "gating by direct coupling to ER depletion". Conceptually, this means opening of the channel by a mechanism independent of other (mainly PLC/IP3/Ca2+) processes. Meanwhile it is rather clear and commonly agreed that most, if not all TRPCs, can be activated by other alternative mechanisms involving Ca2+, G proteins, lipid mediators in a fairly ER depletion-independent manner. Of course, these processes are tightly linked to (especially IP3R/RyR-mediated) store depletion events, and there is indeed some evidence of a potential coupling between TRPCs and ER resident proteins might be relevant. However, there is certainly no consensus in the Ca2+ signaling community that would, at any rate, justify the classification of TRPCs as "gated by store depletion". Especially when it comes to characterization (identification) of SOCE in native tissues such as cardiac muscle, the size of currents, lack of specificity of tools and activation protocols simply do not (or hardly) allow for reliable conclusions about classical SOCE mechanisms/components. I feel that such important conceptual aspects are increasingly obscured in in reports focusing on native tissues and pathophysiology. For instance, considering TRPC3/6/7 mediated cardiac remodeling, there is a quite strong evidence that this phenomenon is most probably due to GPCR/PLC/DAG activation pathways rather than classical SOCE. Please tune down the parts suggesting such signaling mechanisms as SOCE. This may be at the best SOCE related.
Unfortunately, there is an increasing amount of confirmatory reports in the literature that simply refer to previous reports and claim to confirm the hypothesis of native TRPCs operating as “SOCE” based on inconclusive data. I see that this drives a conceptionally misleading drift away from the consensus, which has been achieved at the level of the molecular mechanistic analysis of Ca2+ entry mechanisms and TRPC and STIM/Orai functions. I would ask the authors to include a more critical view on this aspect in their review. Such a critical, differentiated statement is definitely needed up front when talking about generality of SOCE.
Reviewer 2 Report
The manuscript by Sabourin et al. reviewed the literature on the SOCE machinery in left and right ventricular pathophysiology. This review is well performed and interested. However, there are some minor issues that should be considered when revising the manuscript.
1. The authors are suggested to draw a summary diagram.
2. The authors are suggested to discuss SOCE inhibitors or agonists that were used in clinical trial and possible therapeutic applications
Round 2
Reviewer 1 Report
The authors did a great job in amending the manuscript. This revised version is valuable and inspiring review.